# Intensive vital signs monitoring reduces 30-day mortality among stroke patients: A cohort study from Tanzania

Basil Tumaini[1‡]*, Ipyana Kunjumu[1‡], Mohamed Mnacho[2], Patricia Munseri[1]

1 Department of Internal Medicine, Muhimbili University of Health and Allied Sciences, Dar es Salaam, Tanzania, 2 Neurology Unit, Department of Internal Medicine, Muhimbili National Hospital, Dar es Salaam, Tanzania

‡ These authors share first authorship on this work.
* tumainibasil@gmail.com

## Abstract

### Background

Stroke is a leading cause of mortality and disability worldwide, with disproportionate impacts in low- and middle-income countries due to limited healthcare resources. Monitoring vital signs during the acute phase of a stroke is critical for detecting complications. We aimed to assess the impact of intensive (6-hourly) versus usual practice (12-hourly) vital sign monitoring strategies on stroke outcomes at Mloganzila Hospital, Tanzania.

### Methods

We conducted a prospective cohort study of patients with first-ever stroke admitted to Mloganzila Hospital between October 2023 and March 2024. Participants were assigned to 6-hourly or 12-hourly vital signs monitoring for the first 72 hours, after which all received usual care. Monitoring frequency reflected routine clinical practice; no investigational treatments were introduced, and therapeutic decisions were unaffected by group allocation. The primary outcome was 30-day all-cause mortality; secondary outcomes included functional disability measured by the modified Rankin Scale. Ethical approval was obtained, and informed consent was secured. Data were analyzed using SPSS version 26 and Stata version 15. Survival was assessed using Kaplan-Meier analysis, and factors associated with 30-day mortality identified using modified Poisson regression.

### Results

Of 306 participants (mean age: 59.6 ± 14.0 years; 53.6% male), 63.7% experienced a hemorrhagic stroke. There were no significant differences in stroke severity or functional disability at 72 hours between the groups. Intensive vital signs monitoring was

**Data availability statement:** The dataset underlying this study contains sensitive personal and health-related information from a small number of participants at a single center. Although data were de-identified, there remains a risk of re-identification, particularly in a resource-limited setting. Public sharing of the dataset was not approved by the Muhimbili University of Health and Allied Sciences (MUHAS) Institutional Review Board (IRB) at the time of ethics approval. However, the authors support data transparency and reproducibility. Qualified researchers may request access to the dataset through either of the following contacts: 1. Corresponding Author Dr. Basil Tumaini Email: tumainibasil@gmail.com Tel: +255 767 847 258 2. Director of Research and Publications, MUHAS (IRB Contact) P.O. Box 65001, Dar es Salaam, Tanzania Tel: +255-22-2152489 Email: drp@muhas.ac.tz All requests will be reviewed in accordance with MUHAS policies and ethical guidelines.

**Funding:** The author(s) received no specific funding for this work.

**Competing interests:** The authors have declared that no competing interests exist.

associated with significantly lower 30-day mortality compared to usual monitoring (35.9% vs. 49.7%; p = 0.015).

## Conclusion

Frequent monitoring of vital signs during the acute phase of stroke may enhance clinical stability and improve patient outcomes. Our findings demonstrate a significant reduction in 30-day mortality with intensive monitoring. These findings support the establishment of dedicated stroke units with structured monitoring protocols in resource-limited settings to strengthen stroke care systems and reduce preventable deaths.

## Introduction

Stroke remains a leading cause of mortality and disability globally, posing a significant public health burden [1]. With an estimated annual global incidence of 12.2 million cases, it ranks as the second most common cause of death and a major contributor to long-term disability [1]. Low- and middle-income countries (LMICs), including Tanzania, bear a disproportionate share of this burden due to limited access to specialized care and healthcare resources [2–4].

In Tanzania, the stroke burden is escalating, driven by rising rates of hypertension, diabetes, and other modifiable risk factors [2,5]. Yet, the healthcare system remains inadequately prepared for comprehensive stroke care, with most hospitals lacking dedicated stroke units and the capacity for continuous monitoring [2–4]. Fluctuations in vital signs —particularly blood pressure, heart rate, respiratory rate, oxygen saturation, temperature, and blood glucose—during the acute phase of stroke (within the first 72 hours of admission) are critical indicators of hemodynamic instability and impending complications such as recurrent stroke, cerebral edema, and elevated intracranial pressure [6,7]. These derangements are known to influence key outcomes, including disability and mortality.

In the absence of stroke units in Tanzanian hospitals [3], frequent monitoring of vital signs may serve as a pragmatic and cost-effective approach to enhance early complication detection, optimize clinical response, and improve outcomes. While evidence from high-income countries supports the benefits of intensive monitoring in reducing mortality [6,8], such data are scarce in LMICs. This study evaluated the impact of intensive (6-hourly) versus usual practice (12-hourly) vital sign monitoring on 30-day mortality and functional disability among stroke patients admitted to Mloganzila Hospital.

The results of this study will contribute evidence-based recommendations for optimizing stroke care in LMICs.

## Materials and methods

### Ethics statement

This study was approved by the Research and Publication Committee of Muhimbili University of Health and Allied Sciences (approval reference:

DA.282/298.01.C/1935). As the study was not a randomized controlled trial and did not involve external collaborators, additional ethical clearance from the National Institute for Medical Research (NIMR) was not required under current national guidelines. Written informed consent was obtained from all participants or their legal representatives (for patients unable to provide informed consent because of stroke severity). The study adhered to the Declaration of Helsinki, ensuring participant rights, safety, and confidentiality. Data collection was anonymized, with secure storage in locked cabinets and password-protected electronic formats. Analysis was conducted on de-identified datasets. Any abnormal clinical findings were promptly reported to attending clinicians for appropriate management, per hospital protocols.

## Study design

A prospective cohort study was conducted at Mloganzila Hospital, a tertiary referral center in Tanzania, between October 16, 2023, and March 31, 2024. The study evaluated the impact of intensive (6-hourly) versus usual practice at the center (12-hourly) vital signs monitoring on stroke outcomes within 30 days post-admission. At the time of the study, stroke centers were not available in Tanzania [6,9]. We evaluated whether more frequent monitoring, such as 6-hourly, could provide benefits in stroke outcomes even in centers without continuous monitors.

Although random allocation was applied to minimize selection bias, the study was observational in nature. The assigned monitoring schedules reflected existing hospital practices and did not involve investigator-initiated therapeutic interventions, blinding, or experimental treatments. As such, this study is best classified as a prospective cohort study rather than a randomized clinical trial.

## Study population

The study included patients aged ≥18 years, admitted with a clinical diagnosis of first-ever stroke, based on the updated definition of stroke by the American Heart Association/American Stroke Association [10]. Patients with stroke mimics—defined as clinical syndromes initially suspected to be stroke but subsequently diagnosed as other conditions (e.g., brain tumors, metabolic encephalopathies)—were excluded. Patients unable to communicate and without a reliable informant were also excluded. A reliable informant was defined as a relative or caregiver capable of providing accurate clinical and behavioral history for patients with aphasia or impaired consciousness.

## Sample size

The sample size was calculated using a formula for comparing two proportions, a method appropriate for both randomized clinical trials and prospective cohort studies. The aim was to detect a relative reduction in 30-day mortality from 61% in the 12-hourly monitoring group to 37% in the 6-hourly monitoring group These estimates were based on mortality data from two published prospective cohort studies in sub-Saharan Africa: one conducted in Tanzania [2] reporting a 61% mortality rate, and another in Nigeria [11] reporting a 37% mortality rate among stroke patients.

The formula used was: $n = [(Z\alpha/2 + Z\beta)^2 \times (p_1(1-p_1) + p_2(1-p_2))] / (p_1 - p_2)^2$,
where: $p_1 = 0.61$ (expected mortality in the 12-hourly group), $p_2 = 0.37$ (expected mortality in the 6-hourly group), $Z\alpha/2 = 2.576$ (for two-sided $\alpha = 0.01$), and $Z\beta = 1.28$ (for 90% power).

Substituting these values yielded a required sample size of approximately 122 participants per group, or 244 participants in total.

To ensure adequate power despite potential data loss, the sample size was adjusted upward to account for a 10% anticipated non-response or attrition rate, which is common in prospective follow-up studies of acute stroke. To further enhance reliability and allow balanced randomization, the final target sample size was rounded up to 306 participants (153 per group), maintaining ≥90% power to detect the prespecified mortality difference. Participants were randomized into intensive or usual practice monitoring groups in a 1:1 ratio.

## Randomization and allocation

Participants were enrolled upon admission to the general medical wards after stroke diagnosis was confirmed by brain imaging (CT or MRI). Immediately after admission, participants were assigned to one of the two monitoring groups using computer-generated randomization. Allocation was concealed in sealed opaque envelopes, which were opened by a research assistant in the presence of the participant or their next of kin. This ensured proper allocation without participant involvement in the randomization process. Blinding was not implemented in this study. Due to the nature of the monitoring schedules—documented in patient charts—both clinical staff and outcome assessors were necessarily aware of group allocation. All assessments and follow-up procedures were conducted using standardized tools to minimize bias and ensure consistency in data collection.

## Exposure

Vital signs monitoring commenced within 30 minutes of allocation. Intensive monitoring involved measuring vital signs (blood pressure, heart rate, respiratory rate, oxygen saturation, and temperature) every 6 hours for the first 72 hours post-admission. Usual practice monitoring measured the same parameters every 12 hours. All participants received usual practice monitoring after the initial 72 hours until discharge.

## Data collection

**Sociodemographic and comorbidity data.** Trained research assistants and the principal investigator used structured questionnaires to collect sociodemographic data (age, sex, cigarette smoking, and alcohol intake) and comorbidity data (hypertension, diabetes mellitus, chronic kidney disease, and heart disease).

**Clinical data collection.** Vital signs and other clinical parameters were measured as follows:

**Blood pressure (BP):** Measured in the unaffected upper limb using a digital sphygmomanometer following the procedure described in the ESC/ESH guidelines for the management of arterial hypertension [12]. Hypertension was defined as systolic BP ≥ 140 mmHg or diastolic BP ≥ 90 mmHg.

**Pulse rate (PR) and oxygen saturation (SpO$_2$):** Assessed with a fingertip pulse oximeter.

**Respiratory rate (RR):** Counted manually over one minute.

**Temperature:** Measured using a digital thermometer.

**Capillary blood glucose:** Measured using the GLUCOPLUS™ glucometer. Hyperglycemia was defined as glucose levels >8 mmol/L, and hypoglycemia as levels <3.9 mmol/L, in line with American Diabetes Association guidelines [13].

Out-of-range vital sign parameters were promptly communicated to attending clinicians for management, according to the hospital protocol. For patients with hemorrhagic stroke, antihypertensive therapy was initiated if the systolic blood pressure (SBP) exceeded 160 mmHg, with the goal of lowering SBP to <140 mmHg within one hour. In cases where the initial SBP was > 210 mmHg, the BP was lowered by approximately 60 mmHg. Blood pressure reduction was achieved primarily through intravenous antihypertensives (labetalol or hydralazine) to ensure rapid and controlled BP lowering and maintained with oral agents. For patients with ischemic stroke, as thrombolysis was unavailable, oral antihypertensive treatment was initiated if SBP exceeded 220 mmHg or diastolic blood pressure exceeded 120 mmHg, aiming to achieve a 15% reduction in BP over the first 24 hours. If BP was below these thresholds, treatment was generally deferred during the acute phase to preserve cerebral perfusion.

## Laboratory investigations

Venous blood samples (15 mL) were collected aseptically on admission for analysis. Hematological and biochemical parameters were measured using the Hemolyzer 3 Pro and BioSystems A15 analyzers, respectively.

For complete blood counts, hemoglobin levels were categorized as follows: low hemoglobin as <13.2 g/dL in males and <11.5 g/dL in females; high hemoglobin as >16.5 g/dL in males and >16.0 g/dL in females; leukocytosis as a total white blood cell count >11 × 10⁹/L or an absolute neutrophil count >7.5 × 10⁹/L. A platelet count <150 × 10⁹/L was considered thrombocytopenia [14].

For electrolytes: sodium levels were categorized as hyponatremia: mild (130–134 mmol/L), moderate (121–129 mmol/L), severe (<120 mmol/L); or hypernatremia: mild (146–150 mmol/L), moderate (151–170 mmol/L), severe (>170 mmol/L). Potassium levels were categorized as hypokalemia: mild (3.1–3.4 mmol/L), moderate (2.6–3.0 mmol/L), severe (≤2.5 mmol/L); or hyperkalemia: mild (5.1–5.9 mmol/L), moderate (6.0–6.4 mmol/L), and (>6.5 mmol/L) as severe [15].

For the lipid profile: low-density lipoprotein (LDL) cholesterol: elevated levels were defined as ≥100 mg/dL. High-density lipoprotein (HDL) cholesterol: low HDL was defined as <40 mg/dL in males and <50 mg/dL in females. Total cholesterol: hypercholesterolemia was defined as ≥200 mg/dL (≥5.17 mmol/L). Triglycerides: elevated levels were defined as ≥150 mg/dL [16].

All laboratory analyses followed standard operating procedures, with routine quality control checks ensuring the reliability and accuracy of results. Out-of-range laboratory parameters were promptly communicated to attending clinicians for management, according to the hospital protocol.

## Imaging

All participants underwent brain imaging to confirm stroke diagnosis and classify stroke types. Non-contrast computed tomography (CT) scans were acquired using a GE Healthcare Optima scanner, and magnetic resonance imaging (MRI) was performed using a GE SIGNA Creator 1.5T scanner. These imaging procedures were conducted in the Mloganzila Hospital radiology department, following established protocols to ensure consistent and accurate results. A certified radiologist interpreted the images to differentiate between ischemic and hemorrhagic stroke.

## Outcome assessment

Stroke severity was assessed using the National Institutes of Health Stroke Scale [17], and functional outcomes were evaluated with the modified Rankin Scale [18] at 14 and 30 days post-admission by the principal investigator and four medical doctors trained and certified to use the scales. All stroke patients were retained in the hospital for at least 72 hours post-admission. Follow-up at 14 and 30 days was conducted through scheduled in-person visits at the neurology clinic, in line with clinic protocol. All surviving patients attended the clinic for follow-up. For participants who did not attend, phone calls were made to confirm outcomes, including deaths, as reported by relatives or next of kin. The primary outcome was 30-day mortality. Secondary outcomes included stroke severity and functional disability at 14 and 30 days (see S1 Appendix). For analysis purposes, the modified Rankin Scale (mRS) scores of 1 and 2 were combined to represent "mild disability", indicating functional independence. Mortality and disability were compared between the two monitoring groups using Kaplan-Meier survival analysis. Follow-up was complete, and there was no missing data.

## Statistical analysis

Data were analyzed using IBM® SPSS® Statistics version 26 and Stata® version 15. Descriptive statistics summarized demographic, clinical, and laboratory characteristics. Categorical variables were presented as frequencies and percentages and compared between monitoring groups using the chi-square test or Fisher's exact test, as appropriate. Continuous variables were summarized using means and standard deviations or medians and interquartile ranges, and compared using the Student's t-test or Mann-Whitney U test, based on distribution. Kaplan-Meier curves were used to compare 30-day survival by monitoring strategy. Modified Poisson regression was employed to identify independent predictors of 30-day mortality, given the common occurrence of this outcome. Statistical significance was defined as $p < 0.05$.

## Results

A total of 312 participants with a clinical diagnosis of first-ever stroke were screened for eligibility. Six participants were excluded due to stroke mimics (two cases: high-grade glioma and meningioma) or refusal to consent, resulting in 306 participants randomized equally into two groups: 153 participants received intensive (6-hourly) vital signs monitoring, while 153 participants were assigned to usual practice (12-hourly) monitoring (Fig 1).

### Sociodemographic and clinical characteristics of study participants

Baseline sociodemographic and clinical characteristics were mostly comparable between the two groups. The mean age was 59.6 ± 14.0 years, with a predominance of males (164/306, 53.6%). Hemorrhagic strokes accounted for 195 of 306 (63.7%), and 77.8% (238/306) of participants were hypertensive. Cigarette smoking (17.7% vs. 9.1%, p = 0.029) and diabetes mellitus (20.3% vs. 11.8%, p = 0.043) were significantly more prevalent in the intensive monitoring group (Table 1).

### Baseline laboratory findings of study participants

Laboratory results revealed anemia in 113 (36.9%) of the 306 study participants, with thrombocytopenia significantly more prevalent in the intensive monitoring group (20.3% vs. 10.5%; p = 0.017). Electrolyte imbalances, such as hyperkalemia and hyponatremia, were evenly distributed between groups, as were glomerular filtration rate (GFR) categories and the lipid profiles (Table 2).

### Vital signs during the first 72 hours post-admission by monitoring strategy among study participants

At 72 hours post-admission, vital signs were largely comparable between participants in the intensive (6-hourly) and usual practice (12-hourly) monitoring groups. Median systolic blood pressure was similar across groups at 140 mm Hg (interquartile range [IQR]: 130–150 vs. 130–155; p = 0.496). In contrast, diastolic blood pressure was significantly lower in the intensive monitoring group (median: 88 mm Hg; IQR: 80–92) compared to the usual practice group (median: 90 mm Hg; IQR: 85–95; p = 0.004). A total of 80 participants in the intensive monitoring group triggered BP alerts, compared to 72

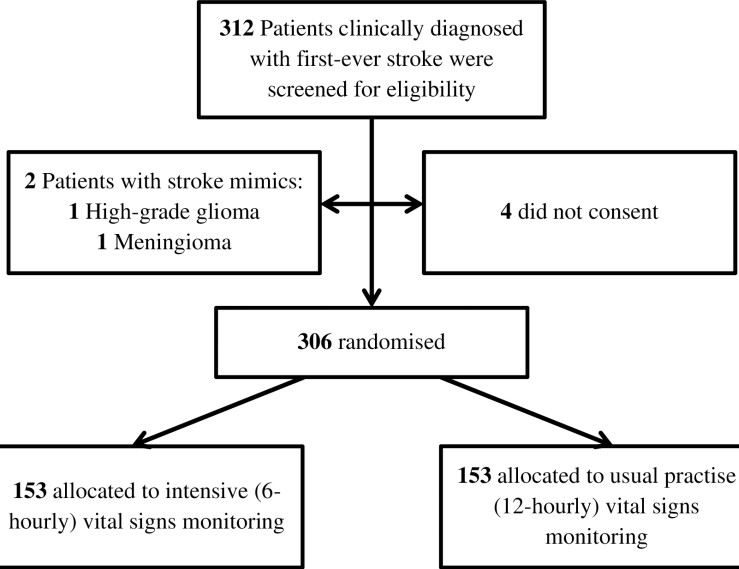

**Fig 1. Recruitment Flowchart.**

**Table 1. Baseline sociodemographic, behavioral, and clinical characteristics of study participants by vital signs monitoring strategy.**

| Characteristic | Total N = 306 | Vital signs monitoring strategy | | p value |
| --- | --- | --- | --- | --- |
| | | 6-hourly n = 153 (50.0%) | 12-hourly n = 153 (50.0%) | |
| Mean age ± SD (years) | 59.6 ± 14.0 | 59.9 ± 14.4 | 59.3 ± 13.5 | 0.686 |
| Male sex | 164 (53.6%) | 85 (55.6) | 79 (51.6) | 0.492 |
| Cigarette smoking | 41 (13.4%) | 27 (17.7) | 14 (9.1) | 0.029 |
| Alcohol consumption | 72 (23.5%) | 33 (21.6) | 39 (25.5) | 0.419 |
| Hypertension | 238 (77.8%) | 122 (79.7) | 116 (75.8) | 0.409 |
| Diabetes mellitus | 49 (16.0%) | 31 (20.3) | 18 (11.8) | 0.043 |
| HIV infection | 19 (6.2%) | 6 (3.9) | 13 (8.5) | 0.097 |
| Chronic kidney disease | 20 (6.5%) | 9 (5.9) | 11 (7.2) | 0.644 |
| Heart disease | 17 (5.6%) | 11 (7.2) | 6 (3.9) | 0.212 |
| **Stroke type** | | | | |
| Hemorrhagic | 195 (63.7%) | 92 (60.1) | 103 (67.3) | 0.191 |
| Ischemic | 111 (36.3%) | 61 (39.9) | 50 (32.7) | |
| **Median (IQR) vital signs** | | | | |
| Systolic BP (mm Hg) | | 162 (143-199) | 159 (141-182) | 0.647 |
| Diastolic BP (mm Hg) | | 97 (87-110) | 97 (86-115) | 1.000 |
| Heart rate (bpm) | | 85 (72-96) | 89 (78-100) | 0.170 |
| Respiratory rate (cycles/min) | | 19 (18-21) | 20 (18-22) | 0.360 |
| SpO$_2$ (%) | | 98 (97-99) | 98 (97-99) | 0.328 |
| Temperature (°C) | | 36.7 (36.4-37.0) | 36.6 (36.4-36.8) | 0.360 |
| Blood glucose (mmol/L) | | 7.2 (6.1-9.2) | 7.1 (5.9-8.3) | 0.568 |
| **NIHSS score** | | | | |
| Minor stroke | 26 (8.5) | 10 (6.5) | 16 (10.5) | |
| Moderate stroke | 156 (51.0) | 76 (49.7) | 80 (52.3) | 0.258 |
| Moderate to severe stroke | 66 (21.6) | 32 (20.9) | 34 (22.2) | |
| Severe stroke | 58 (19.0) | 35 (22.9) | 23 (15.0) | |

bpm – beats per minute; HIV – Human Immunodeficiency Virus; IQR – interquartile range; NIHSS – National Institutes of Health Stroke Scale; SD – standard deviation; SpO$_2$ - peripheral capillary oxygen saturation

participants in the usual monitoring group; this difference was not statistically significant (p = 0.457). Other clinical parameters, including heart rate, respiratory rate, oxygen saturation (SpO$_2$), body temperature, and capillary blood glucose levels, did not differ significantly between the groups (Table 3 and S1 Fig).

### Stroke severity and disability at 72 hours among study participants

At 72 hours post-admission, stroke severity, assessed by the National Institutes of Health Stroke Scale (NIHSS) score, showed no significant difference between the groups. Severe strokes were observed in 34 (22.2%) of 153 participants in the intensive group compared to 25 of 153 (16.3%) in the usual practice group (p = 0.565 for difference in stroke severity between the two vital signs monitoring groups). Functional disability also showed comparable trends between groups (Table 4).

### Outcomes at 14 days among study participants

At 14 days, functional disability assessed by the modified Rankin Scale (mRS) revealed no significant difference between groups. The proportion of participants with mild disability was 9/153 (5.9%) in the intensive group and 12/153 (7.8%) in the

**Table 2. Baseline laboratory characteristics of study participants by vital signs monitoring strategy.**

| Characteristic | Total N = 306 | Vital signs monitoring strategy | | p value |
| --- | --- | --- | --- | --- |
| | | 6-hourly n = 153 (50.0%) | 12-hourly n = 153 (50.0%) | |
| **Complete blood count** | | | | |
| Anemia | 113 (36.9%) | 52 (34.0%) | 61 (39.9%) | 0.286 |
| Leukocytosis | 133 (43.5%) | 65 (42.5%) | 68 (44.4%) | 0.729 |
| Thrombocytopenia | 47 (15.4%) | 31 (20.3%) | 16 (10.5%) | 0.017 |
| **Electrolytes** | | | | |
| Hyponatremia | 73 (23.9%) | 36 (23.5%) | 37 (24.2%) | |
| Hypernatremia | 42 (13.7%) | 23 (15.0%) | 19 (12.4%) | |
| Hypokalemia | 69 (8.5%) | 31 (20.3) | 38 (24.8) | |
| Hyperkalemia | 26 (8.5%) | 16 (10.5%) | 10 (6.5%) | |
| **GFR categories** | | | | |
| G1 | 63 (20.6%) | 33 (21.6%) | 30 (19.6%) | |
| G2 | 105 (34.3%) | 54 (35.3%) | 51 (33.3%) | |
| G3A | 67 (21.9%) | 33 (21.6%) | 34 (22.2%) | 0.761 |
| G3B | 30 (9.8%) | 14 (9.2%) | 16 (10.5%) | |
| G4 | 21 (6.9%) | 12 (7.8%) | 9 (5.9%) | |
| G5 | 20 (6.5%) | 7 (4.6%0 | 13 (8.5%) | |
| **Lipid profile** | | | | |
| Hypercholesterolemia | 88 (34.6%) | 46 (37.4%) | 42 (32.1%) | 0.372 |
| Hypertriglyceridemia | 57 (22.4%) | 28 (22.6%) | 29 (22.1%) | 0.932 |

GFR: glomerular filtration rate.

**Table 3. Vital signs at 72 hours by monitoring strategy among study participants.**

| Characteristic | Total N = 306 | Vital signs monitoring strategy | | p value |
| --- | --- | --- | --- | --- |
| | | 6-hourly n = 153 (50.0%) | 12-hourly n = 153 (50.0%) | |
| **Median (IQR) vital signs** | | | | |
| Systolic BP (mm Hg) | | 140 (130-150) | 140 (130-155) | 0.496 |
| Diastolic BP (mm Hg) | | 88 (80-92) | 90 (85-95) | 0.004 |
| Heart rate (bpm) | | 82 (76-89) | 82 (77-91) | 0.440 |
| Respiratory rate (cycles/min) | | 19 (18-20) | 19 (18-20) | 0.062 |
| SpO$_2$ (%) | | 98 (97-99) | 98 (97-99) | 0.497 |
| Temperature (°C) | | 36.5 (36.2-36.7) | 36.5 (36.2-36.7) | 0.979 |
| Blood glucose (mmol/L) | | 5.9 (5.2-6.9) | 6.0 (5.2-7.1) | 0.948 |

bpm – beats per minute; IQR – interquartile range; SpO$_2$ - peripheral capillary oxygen saturation

usual practice group (p = 0.848). Mortality rates were also comparable: 39/153 (25.5%) versus 37/153 (24.2%), as shown in S1 Table.

## Thirty-day mortality and functional outcomes among study participants

At 30 days, mortality was significantly lower in the intensive monitoring group (55/153, 35.9%) compared with the usual practice group (76/153, 49.7%; p = 0.015). Kaplan-Meier survival analysis demonstrated better survival in the intensive

**Table 4. Stroke severity and functional disability at 72 hours among study participants by vital signs monitoring strategy.**

| Characteristic | Total N = 306 | Vital signs monitoring strategy | | p value |
| --- | --- | --- | --- | --- |
| | | 6-hourly n = 153 (50.0%) | 12-hourly n = 153 (50.0%) | |
| **Stroke severity at 72 hours** | | | | |
| Minor | 29 (9.5) | 13 (8.5) | 16 (10.5) | |
| Moderate | 145 (47.4) | 69 (45.1) | 76 (49.7) | 0.565 |
| Moderate to severe | 73 (23.9) | 37 (24.2) | 36 (23.5) | |
| Severe | 59 (19.3) | 34 (22.2) | 25 (16.3) | |
| **Functional disability at 72 hours** | | | | |
| No significant disability | 7 (2.3) | 3 (2.0) | 4 (2.6) | |
| Slight disability | 10 (3.3) | 4 (2.6) | 6 (3.9) | |
| Moderate disability | 36 (11.8) | 15 (9.8) | 21 (13.7) | 0.761 |
| Moderate to severe disability | 145 (47.4) | 75 (49.0) | 70 (45.8) | |
| Severe disability | 108 (35.3) | 56 (36.6) | 52 (34.0) | |

monitoring group (log-rank p = 0.022), see Fig 2. Functional outcomes showed no statistically significant difference between the two groups in terms of severe disability (20.9% vs. 18.3%; p = 0.178), as shown in Table 5.

### Independent predictors of 30-day mortality

To address potential confounding factors influencing 30-day mortality, we conducted a multivariable analysis using modified Poisson regression, as mortality was a common outcome and logistic regression could overestimate associations.

As shown in S2 Table, baseline stroke severity (NIHSS ≥16) was the strongest independent predictor of 30-day mortality (relative risk [RR]: 5.14, 95% CI: 3.71–7.11; p < 0.001). Other significant predictors included current alcohol use (RR: 1.40, 95% CI: 1.12–1.74; p = 0.003), heart disease (RR: 1.60, 95% CI: 1.07–2.37; p = 0.020), and age ≥60 years (RR: 1.25, 95% CI: 1.00–1.57; p = 0.049).

Importantly, intensive vital signs monitoring remained independently associated with a 35% reduction in the risk of 30-day mortality (RR: 0.65, 95% CI: 0.53–0.79; p < 0.001), after adjustment for age, sex, behavioral risk factors, comorbidities, stroke type, and stroke severity.

### Discussion

This study underscores the significant impact of intensive vital signs monitoring on improving stroke outcomes, particularly in resource-limited settings. By systematically analyzing the effects of monitoring strategies on vital signs, disability levels, and mortality, our findings provide critical insights into optimizing stroke care.

Vital signs represent critical markers of physiological status in the acute phase of stroke and are central to early clinical decision-making. In this study, most parameters at 72 hours post-admission—including systolic blood pressure, heart rate, respiratory rate, oxygen saturation, temperature, and blood glucose—did not significantly differ between monitoring groups. However, diastolic blood pressure was modestly but significantly lower in the intensive (6-hourly) monitoring group compared to the usual practice monitoring (12-hourly) group. While this difference is noteworthy, it should be interpreted as an association rather than a direct consequence of the monitoring strategy. Given the randomized design, both groups were expected to be comparable at baseline. The observed difference may reflect evolving hemodynamic responses influenced by more timely clinical recognition and management in the intensively monitored group. Structured monitoring may have enabled earlier detection of deviations from normal physiology and prompted prompt intervention, consistent with findings from stroke unit care models in high-income settings [6,9].

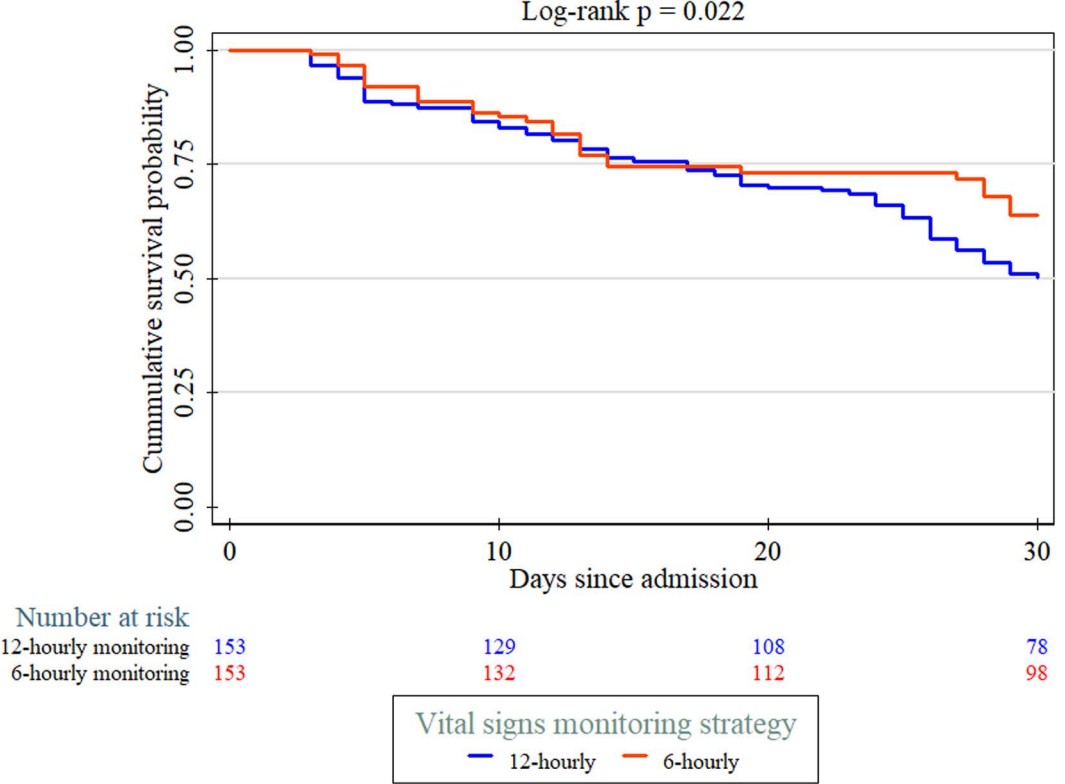

**Fig 2. Kaplan-Meier survival analysis of 30-day mortality among stroke patients, comparing intensive (6-hourly) vs. usual (12-hourly) vital signs monitoring.** Intensive monitoring improved survival (log-rank p = 0.022).

**Table 5. Functional disability and mortality at 30 days among stroke patients by vital signs monitoring strategy.**

| Outcome | Total N = 306 | Vital signs monitoring strategy | | p value |
|---|---|---|---|---|
| | | 6-hourly n = 153 (50.0%) | 12-hourly n = 153 (50.0%) | |
| **Functional disability by mRS** | | | | |
| Mild disability | 23 (7.5%) | 14 (9.2%) | 9 (5.9%) | |
| Moderate disability | 34 (11.1%) | 19 (12.4%) | 15 (9.8%) | 0.178 |
| Moderate to severe disability | 58 (19.0%) | 33 (21.6%) | 25 (16.3%) | |
| Severe disability | 60 (19.6%) | 32 (20.9%) | 28 (18.3%) | |
| **Mortality at 30 days** | | | | |
| Yes | 131 (42.8%) | 55 (35.9%) | 76 (49.7%) | 0.015 |
| No | 175 (57.2%) | 98 (64.1%) | 77 (50.3%) | |

mRS: the modified Rankin Scale

The value of frequent vital signs monitoring in acute stroke care lies not in altering physiological parameters directly, but in enabling more responsive and informed care. In resource-limited settings without continuous monitoring technology, scheduled 6-hourly assessments offer a feasible and impactful strategy to detect early deterioration. These findings support the inclusion of structured monitoring protocols in emerging stroke care pathways, particularly where stroke units have yet to be established.

Functional disability, assessed using the mRS, was substantial in the early phase of stroke care. At 72 hours post-admission, nearly half of the participants exhibited moderate to severe disability, and over one-third were severely disabled. Notably, these rates were similar across the two monitoring groups, indicating that while intensive vital signs monitoring may facilitate earlier recognition and stabilization of acute physiological derangements, it did not translate into measurable differences in early functional recovery. The lack of reperfusion strategies for acute ischemic stroke at the study site may also have limited the potential for early neurological improvement.

However, by 30 days post-admission, the burden of disability had declined considerably, with moderate to severe disability decreasing to 19.0% and severe disability to 19.6%. This improvement reflects the expected trajectory of post-stroke neurological recovery over time, likely supported by early clinical stabilization and access to rehabilitation services. Although a formal stroke unit was not available, inpatient and outpatient rehabilitation services were accessible and routinely prescribed at discharge, ensuring that most survivors received some degree of post-acute rehabilitative care.

These findings align with global observations from high-income countries, where functional recovery typically progresses over the first month, particularly with early access to structured rehabilitation programs [19]. The parallels highlight the potential for significant recovery even in resource-constrained settings, provided there is timely stabilization, continuity of care, and access to basic rehabilitation. The integration of formal stroke units with embedded rehabilitation pathways remains a critical goal to further improve functional outcomes in Tanzania and similar low-resource contexts [20].

A key finding of this study is the significant reduction in 30-day mortality among patients receiving intensive monitoring. The observed mortality of 35.9% in the intensive group represents a marked improvement compared to the 49.7% mortality in the usual practice monitoring group. These results align with global evidence suggesting that early detection and intervention for clinical deterioration are critical in improving survival rates [6,8,9,21]. Given the multifactorial nature of stroke outcomes, we undertook an adjusted analysis using multivariable modified Poisson regression, an approach appropriate for outcomes with high prevalence such as 30-day mortality. This adjustment accounted for potential confounders, including sociodemographic characteristics, behavioral risk factors, comorbid conditions, stroke type, and baseline stroke severity. Notably, intensive vital signs monitoring remained independently associated with a substantial 35% reduction in the risk of 30-day mortality after adjustment. These findings underscore the robustness of the observed association and support the implementation of intensive monitoring strategies as a pragmatic intervention to improve stroke outcomes in resource-limited settings.[3,22]

The findings advocate for the establishment of dedicated stroke units in Tanzania, characterized by multidisciplinary teams and continuous monitoring. Beyond reducing mortality, such units could serve as hubs for early rehabilitation, improving both survival and quality of life for stroke patients. Policymakers must prioritize investments in infrastructure, training, and staffing to operationalize these units. Referral systems should also be optimized to address inequities in stroke care access and ensure timely intervention for severe cases.

### Limitations and future directions

This study has several limitations that warrant consideration. First, vital signs were measured using manual techniques, which may have introduced variability in measurements. Although staff were trained and standard protocols were followed, automated monitoring systems may offer greater precision and should be explored in future studies. Second, while participants were randomized and baseline stroke severity was comparable between groups, the intensive monitoring group had a higher prevalence of diabetes mellitus and cigarette smoking—factors independently associated with poorer outcomes. Although these imbalances may have occurred by chance in this moderately sized sample, they raise the possibility of residual confounding that could have influenced the observed outcomes. Third, the high proportion of hemorrhagic strokes in our cohort—likely reflecting referral patterns to our tertiary center—may limit the generalizability of findings to other settings, particularly those with different stroke type distributions. The impact of such referral biases on stroke outcomes should be explored in future multicenter studies.

## Conclusion

This study demonstrates that intensive monitoring of vital signs during the acute phase of stroke significantly reduces 30-day mortality in a resource-limited setting. The findings highlight the critical role of frequent vital sign assessments in mitigating adverse outcomes such as hemodynamic instability, which is closely associated with increased mortality in stroke patients. While functional outcomes at 72 hours and 30 days did not show significant differences between groups, intensive monitoring offers a pragmatic and impactful intervention to improve survival. These results underscore the urgent need for structured protocols and resources for stroke care in Tanzania and similar settings. In particular, they support the establishment of dedicated stroke units equipped with standardized monitoring protocols and integrated rehabilitation services to optimize stroke outcomes and reduce preventable deaths.

## Recommendations

**Establishment of stroke units**: Healthcare systems in Tanzania and other low-resource settings should prioritize the establishment of specialized stroke units equipped with continuous vital sign monitors. These units should include multi-disciplinary teams trained to provide comprehensive stroke care.

**Policy integration**: Incorporate intensive vital sign monitoring protocols, particularly during the first 72 hours post-admission, into national stroke management guidelines to standardize care and improve outcomes.

**Capacity building**: Strengthen the capacity of healthcare facilities by training healthcare professionals in advanced stroke management and ensuring adequate staffing levels to support intensive monitoring.

## Supporting information

**S1 Appendix.  Case report form.**
(PDF)

**S1 Table.  Functional disability and mortality at 14 days among study participants by vital signs monitoring strategy.**
(DOCX)

**S2 Table.  Independent predictors of 30-day mortality among study participants.**
(DOCX)

**S1 Fig.  Blood pressure variability during the first 72 hours post-admission by vital signs monitoring strategy.**
(PDF)

## Acknowledgments

We extend our sincere gratitude to the stroke patients and their families for their participation and trust in this study. Special thanks to the dedicated clinical staff at Mloganzila Hospital for their invaluable support in patient care and data collection. We also appreciate the research assistants whose diligence in data collection and follow-up was instrumental to this work. Finally, we are grateful to our families and colleagues for their unwavering encouragement and support throughout this research journey.

## Author contributions

**Conceptualization:** Ipyana Kunjumu, Mohamed Mnacho, Patricia Munseri.

**Data curation:** Ipyana Kunjumu.

**Formal analysis:** Basil Tumaini, Ipyana Kunjumu.

**Investigation:** Ipyana Kunjumu.

**Methodology:** Basil Tumaini, Ipyana Kunjumu, Patricia Munseri.

**Supervision:** Mohamed Mnacho, Patricia Munseri.

**Writing – original draft:** Basil Tumaini, Ipyana Kunjumu, Mohamed Mnacho, Patricia Munseri.

**Writing – review & editing:** Basil Tumaini, Ipyana Kunjumu, Mohamed Mnacho, Patricia Munseri.

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
