## [Decision Letter · Decision Letter 0]

PONE-D-25-10513Intensive vital signs monitoring reduces 30-day mortality among stroke patients: A cohort study from TanzaniaPLOS ONE

Dear Dr. Tumaini,

Thank you for submitting your manuscript to PLOS ONE. After careful consideration, we feel that it has merit but does not fully meet PLOS ONE’s publication criteria as it currently stands. Therefore, we invite you to submit a revised version of the manuscript that addresses the points raised during the review process.

We look forward to receiving your revised manuscript.

Kind regards,

Elvan Wiyarta, M.D.

Academic Editor

PLOS ONE

Journal Requirements:

Reviewers' comments:

Reviewer's Responses to Questions

**Comments to the Author**

1. Is the manuscript technically sound, and do the data support the conclusions?

Reviewer #1: Partly

Reviewer #2: Partly

Reviewer #3: Partly

2. Has the statistical analysis been performed appropriately and rigorously? 

Reviewer #1: Yes

Reviewer #2: Yes

Reviewer #3: No

3. Have the authors made all data underlying the findings in their manuscript fully available?

Reviewer #1: No

Reviewer #2: Yes

Reviewer #3: Yes

4. Is the manuscript presented in an intelligible fashion and written in standard English?

Reviewer #1: Yes

Reviewer #2: Yes

Reviewer #3: Yes

5. Review Comments to the Author

Reviewer #1: Title: Ok

Abstract: OK, provided the concerns in the methodology section below are addressed and reflected in the abstract in the corrected version

Introduction: OK

Methodology:

Ethics: OK

Study design: A randomized clinical trial would have been a better study design in this case. In fact, according to the descriptions given by the authors of the randomization and allocation of various study arms, it seems that a trial has indeed been conducted. Why the authors are calling it a cohort study is still unclear. The authors need to justify why this is a cohort study, and if that is the case, then why a randomized trial was not conducted.

Study population: ok

Sample size: please elaborate how the sample size was calculated and what formula was used. It seems the formula used is also that of an RCT

Data collection: At what point were patients captured? Were they captured in the emergency department upon admission? were they captured once admitted in the ward? for those in the intervention arm, how soon after being captured did the intensive monitoring start? Was there a difference in care whether a patient who was allocated intensive vs conventional arm eg admission in ICU for any of the groups? is there any justification for excluding other vital paramters from the monitoring eg GCS, NIHSS. Were assessors trained in the assessment of these vital signs?

Outcome assessment: Ok

Statistical analysis: was the chi squared test used in the analysis?

Results:

Sociodempgraphic and clinical characteristics: Baseline sociodemographic and clinical characteristics were NOT comparable between the two groups for cigarette smoking, DM status. any explanation for this? It is also interesting to note that ALL patients in this study were CKD patients. This deserves a special mention and explanation

vital signs at 72 hrs: the observable statistical difference in diastolic BP between the 2 groups may reflect a mere numeric relationship. clinically a diastolic blood pressure of 88mmHg and 90mmHg may have not have a significant biological affect on the outcomes. So it may be statistically significant but with no practical clinical implications.

Stroke severity at 72 hrs; Ok

Outcomes at 14 days: ok

30 day mortality : ok

Discussion: quite brief.

conclusion: ok

Recommendations: OK

referenecs: Ok

the figshare link provided at the end of the manuscript that contains all the raw data is not accessible.

Generally an important topic but important methodological concerns need to be addressed

Reviewer #2: This is a very well written and compelling study that supports the establishment of stroke units with advanced monitoring capabilities in resource-limited settings.

Abstract:

- In the background, there are a lot of parenthesis which is distracting to the reader. Consider removing the parentheses with outcomes as you can list them below in the methods.

- No need to say 306 patients twice, put in results only.

- In the results you mention diastolic blood pressure variability at 72 hours as a main finding but above say you are reporting on outcomes at 30 days. Also, you go beyond simply listing the result and interpret it when you say “underscoring its role in clinical stabilization”; consider removing the interpretation and putting this in the Conclusion instead.

- First sentence of the Conclusion restates the results almost verbatim.

- Excellent final sentence that makes a compelling argument and summarizes the study findings.

Introduction:

- This is extremely well written, concise, and beautifully organized. Well done! Just a few comments to improve it:

- One addition that could be helpful to this section would be listing the vital signs you are referring to. Is it heart rate, blood pressure, oxygen saturation? More?

- Lines 66-68 are basically restating what is above and I don’t think they are needed at all. The introduction ended beautifully at line 65 on the previous page. If you really wanted a final sentence remove the repeated info and leave only “…Hospital. The results of this study will contribute evidence-based recommendations for optimizing stroke care in LMICs.”

Methods:

- Was ethical approval obtained from NIMR, the National IRB of Tanzania, in addition to the Research and Publication Committee of Muhimbili? I believe studies in Tanzania need NIMR approval, too.

- Were the abnormal vitals reported to clinicians in real time? (line 77-78) Consider adding this detail.

- Lines 84-85. You do not need to re-state that studies indicated improved stroke outcomes as you already built the case well in the Introduction. Please remove.

- Line 85. Do not say “we wanted to evaluate” but rather say “we evaluated” as it is more direct, crisp writing.

- Line 91-92. Perhaps define stroke mimics and also reliable informant when describing exclusion criteria.

- In Sample Size, why was 35% relative reduction chosen?

- For the randomization section, why was it opened by the participant? In order to take the vital signs, I’m assuming a healthcare provider had to know if they were randomized to one group vs the other so they could take the vital signs on time, so how was it blinded?

- Line 161. How was follow-up performed? By calling participants or home visits or clinic follow-up visits?

Results:

- Overall excellent job summarizing the pertinent findings from the tables.

- Vitals signs at 72 hours by monitoring strategy – Was the intensive monitoring group just sicker, and that is why they had increased mortality? This should be discussed in Limitations or Discussion.

- There are so many tables and figures. Consider removing Table 5 that talks about outcomes at 14 days, or including it as a supplemental table, since there was nothing significant anyway. The more important results are the 30 day outcomes tables.

- Another way to cut down on the number of tables would be to combine tables 6 and 7 into an outcomes table to include both disability and mortality.

Discussion:

- I haven’t seen title headings for sections of this Discussion before, and don’t really like them. I’ll leave it up to the editor to decide if they will allow the authors to do this or not, but would advise against it.

- Lines 271-272. I think the interpretation here as the vital signs monitoring reduced diastolic blood pressure is incorrect. The monitoring didn’t reduce the blood pressure, but rather it just was that patients that had more intensive monitoring had lower BP. Please reword this inaccurate sentence. This is also where you can discuss if perhaps the intensive monitoring group were just sicker since they had lower diastolic BP. Also consider adding this to the Limitations.

- The whole vital signs monitoring paragraph needs to be re-thought as per above.

- The discussion of functional disability at 72 hours is good, but it doesn’t seem to be highlighted in the Intro/outcomes as a main outcome. Perhaps add it above. Also, consider discussing functional disability at 30 days.

- In the Conclusion paragraph, consider a sentence like that in the abstract that calls for the establishment of stroke units – this was very strong.

Reviewer #3: This study examines the impact of intensive blood pressure (BP) monitoring in LMICs, particularly in settings with limited stroke care resources and without access to a dedicated stroke unit. There are few clarifications and comments are needed:

1. Please consider performing an adjusted analysis to account for potential confounding factors that could influence the main outcome—30-day mortality. This will help determine whether other variables may have affected the results.

2. It would be helpful to present the trend of BP readings over the 72-hour monitoring period (e.g., every 6 or 12 hours). Reporting this can provide insight into BP variability between groups, which is known to be an important factor in stroke complications.

3. If any abnormal BP readings or clinical issues were detected during the monitoring period, please clarify how they were managed. How many patients in each group experienced these issues, and what treatments, if any, were given? what cut-off point value of BP set to start treatment (if needed)

4. Kindly clarify whether BP readings were taken every 6 hours for the full 72 hours or until the patient was discharged, whichever came first.

6. PLOS authors have the option to publish the peer review history of their article (what does this mean? ). If published, this will include your full peer review and any attached files.

**Do you want your identity to be public for this peer review?** For information about this choice, including consent withdrawal, please see our Privacy Policy .

Reviewer #1: No

Reviewer #2: No

Reviewer #3: No

---

## [Author Response · Author response to Decision Letter 1]

13 May 2025

Response to Reviewers’ Comments

Journal Requirements

Response: Thank you. We have reviewed the PLOS ONE formatting guidelines and revised the manuscript to meet the Journal's style requirements.

At this time, please upload the minimal data set necessary to replicate your study's findings to a stable, public repository (such as figshare or Dryad) and provide us with the relevant URLs, DOIs, or accession numbers that may be used to access these data.

Response: We appreciate the journal’s commitment to open science and transparency. In response to your request to upload the minimal dataset to a public repository, we respectfully provide the following revised Data Availability Statement, taking into account institutional policy and ethical constraints:

Data Availability

The dataset underlying this study contains sensitive personal and health-related information from a small number of participants at a single center. Although data were de-identified, there remains a risk of re-identification, particularly in a resource-limited setting with limited patient pools. Public sharing of the dataset was not approved by the Muhimbili University of Health and Allied Sciences (MUHAS) Institutional Review Board at the time of ethics approval.

However, the authors support transparency and reproducibility. Qualified researchers may request access to the dataset by contacting the Chairperson of the MUHAS IRB via the Corresponding Author, Dr. Basil Tumaini (email: tumainibasil@gmail.com), who will facilitate the process in accordance with institutional policies and ethical guidelines.

Reviewer #1:

We thank Reviewer #1 for the constructive and insightful feedback. Below is our detailed point-by-point response:

Title: Ok

Response: Thank you.

Abstract: OK, provided the concerns in the methodology section below are addressed and reflected in the abstract in the corrected version

Response: We agree with the Reviewer. The Abstract has been revised to reflect clarifications in the Methods section, including the justification of the study design.

Introduction: OK

Response: Thank you.

Methodology:

Ethics: OK

Response: Thank you.

Study design: A randomized clinical trial would have been a better study design in this case. In fact, according to the descriptions given by the authors of the randomization and allocation of various study arms, it seems that a trial has indeed been conducted. Why the authors are calling it a cohort study is still unclear. The authors need to justify why this is a cohort study, and if that is the case, then why a randomized trial was not conducted.

Response: We sincerely thank the reviewer for this important observation. We have clarified in the revised manuscript that this was a prospective cohort study, not a randomized clinical trial (RCT). While random allocation was used to strengthen internal validity, the study does not meet key criteria for an RCT for the following reasons:

a. Nature of the exposure: The monitoring schedules (6-hourly vs. 12-hourly vital signs checks) were derived from routine hospital practice and reflected existing standards of care. No investigational treatments, medications, or procedures were introduced by the research team.

b. Lack of blinding and protocol-driven interventions: All healthcare providers and assessors were aware of group allocation, and treatment decisions—including medication, fluids, and investigations—were made independently by clinical teams, without influence from the study protocol.

c. Study intent: The study aimed to evaluate real-world clinical outcomes associated with existing monitoring practices, not to test the efficacy of a novel intervention under controlled conditions. This aligns with the principles of observational research.

d. Ethical and operational considerations: As both monitoring strategies represented acceptable care standards, the study received ethical approval as an observational cohort study embedded within hospital workflows.

We have revised the Study design section of the manuscript accordingly to reflect this clarification and have ensured consistency throughout the text.

Study population: ok

Response: Thank you.

Sample size: please elaborate how the sample size was calculated and what formula was used. It seems the formula used is also that of an RCT

Response: We thank the Reviewer for this important point. The sample size was calculated using a formula for comparing two proportions, a method appropriate for both randomized clinical trials and prospective cohort studies. The aim was to detect a relative reduction in 30-day mortality from 61% in the 12-hourly monitoring group to 37% in the 6-hourly monitoring group These estimates were based on mortality data from two published prospective cohort studies in sub-Saharan Africa: one conducted in Tanzania [1] reporting a 61% mortality rate, and another in Nigeria [2] reporting a 37% mortality rate among stroke patients.

The formula used was: n = [(Zα/2 + Zβ)² × (p₁(1−p₁) + p₂(1−p₂))] / (p₁ − p₂)²,

where: p₁ = 0.61 (expected mortality in the 12-hourly group), p₂ = 0.37 (expected mortality in the 6-hourly group), Zα/2 = 2.576 (for two-sided α = 0.01), Zβ = 1.28 (for 90% power).

We have revised the Sample Size section to include the formula and parameters.

References

1. Matuja SS, Munseri P, Khanbhai K. The burden and outcomes of stroke in young adults at a tertiary hospital in Tanzania: a comparison with older adults. BMC Neurology. 2020;20:1-10. Doi: 10.1186/s12883-020-01793-2

2. Femi OL, Mansur N. Factors associated with death and predictors of one-month mortality from stroke in Kano, Northwestern Nigeria. Journal of neurosciences in rural practice. 2013;4(Suppl 1):S56. Doi: 10.4103/0976-3147.116460

Data collection: At what point were patients captured? Were they captured in the emergency department upon admission? were they captured once admitted in the ward? for those in the intervention arm, how soon after being captured did the intensive monitoring start? Was there a difference in care whether a patient who was allocated intensive vs conventional arm eg admission in ICU for any of the groups? is there any justification for excluding other vital paramters from the monitoring eg GCS, NIHSS. Were assessors trained in the assessment of these vital signs?

Response: We thank the Reviewer for these excellent questions. Clarifications have been made in the Randomization and allocation and Exposure sections of the revised manuscript as follows:

Patients were captured upon admission to the medical ward after stroke diagnosis was confirmed with imaging.

Allocation to either monitoring group was done immediately after admission, and monitoring commenced within 30 minutes of randomization.

All participants were managed in the same general medical wards. None were admitted to ICU, and there was no difference in level of care between groups, aside from the frequency of vital signs monitoring.

NIHSS scores were collected at admission and follow-up but were not recorded as part of the vital signs monitoring schedule. GCS was not routinely used because NIHSS was preferred for stroke severity classification.

All data collectors and outcome assessors were trained medical officers, certified in the use of the NIHSS and mRS.

Outcome assessment: Ok

Response: Thank you.

Statistical analysis: was the chi squared test used in the analysis?

Response: Yes. The chi-square test was used to compare categorical variables including demographic, clinical and laboratory characteristics. This has been explicitly mentioned in the Statistical analysis section in the revised manuscript.

Results:

Sociodempgraphic and clinical characteristics: Baseline sociodemographic and clinical characteristics were NOT comparable between the two groups for cigarette smoking, DM status. any explanation for this?

Response: We agree and have now acknowledged this in the Results and Limitations sections. These imbalances, despite randomization, could be due to chance in a moderately sized sample. We have noted their potential influence on outcomes and addressed this as a limitation.

It is also interesting to note that ALL patients in this study were CKD patients. This deserves a special mention and explanation

Response: We sincerely apologize for the confusion. GFR stages G1 and G2 have now been included in Table 2.

vital signs at 72 hrs: the observable statistical difference in diastolic BP between the 2 groups may reflect a mere numeric relationship. clinically a diastolic blood pressure of 88mmHg and 90mmHg may have not have a significant biological affect on the outcomes. So it may be statistically significant but with no practical clinical implications.

Response: We agree with the Reviewer. In the revised Discussion section, we have clarified that the importance of this finding lies more in its association with survival, rather than in the numerical BP reduction itself. (lines 339-347 in the Revised manuscript with track changes)

Stroke severity at 72 hrs; Ok

Response: Thank you.

Outcomes at 14 days: ok

Response: Thank you.

30 day mortality : ok

Response: Thank you.

Discussion: quite brief.

Response: The Discussion has been expanded to better contextualize the findings, clarify limitations (e.g., imbalanced baseline variables), and discuss implications for stroke care in low-resource settings.

conclusion: ok

Response: Thank you.

Recommendations: OK

Response: Thank you.

referenecs: Ok

Response: Thank you.

the figshare link provided at the end of the manuscript that contains all the raw data is not accessible.

Response: Thank you for highlighting this issue. We have removed the inaccessible figshare link. Due to ethical restrictions and conditions of informed consent, the dataset cannot be publicly shared. However, we have updated the Data Availability section to reflect that data will be made available upon reasonable request to the MUHAS IRB via the Corresponding Author. This ensures compliance with institutional policy while supporting transparency.

Generally an important topic but important methodological concerns need to be addressed

Response: We appreciate the Reviewer’s recognition of the relevance of our study. We have addressed all methodological concerns raised and revised the manuscript to improve clarity, accuracy, and transparency. 

Reviewer #2:

This is a very well written and compelling study that supports the establishment of stroke units with advanced monitoring capabilities in resource-limited settings.

We are grateful to Reviewer #2 for the generous and thoughtful feedback. Your kind words and detailed critique have greatly helped us refine and elevate the clarity, accuracy, and presentation of our work. Below, we address each point raised:

Abstract:

- In the background, there are a lot of parenthesis which is distracting to the reader. Consider removing the parentheses with outcomes as you can list them below in the methods.

Response: We agree. Parentheses have been removed from the Background section of the Abstract, and outcomes are now introduced in a cleaner structure within the Methods section.

- No need to say 306 patients twice, put in results only.

Response: This duplication has been removed. The number of participants is now stated only in the Results subsection.

- In the results you mention diastolic blood pressure variability at 72 hours as a main finding but above say you are reporting on outcomes at 30 days. Also, you go beyond simply listing the result and interpret it when you say “underscoring its role in clinical stabilization”; consider removing the interpretation and putting this in the Conclusion instead.

Response: Thank you for observing this. The phrase “underscoring its role in clinical stabilization” has been removed from the Results subsection of the Abstract and moved appropriately to the Conclusion.

- First sentence of the Conclusion restates the results almost verbatim.

Response: We have revised the first sentence of the Conclusion in the Abstract to avoid repetition and focus on broader interpretation rather than reiterating statistical results.

- Excellent final sentence that makes a compelling argument and summarizes the study findings.

Response: Thank you.

Introduction:

- This is extremely well written, concise, and beautifully organized. Well done!

Response: We are grateful for this encouragement.

Just a few comments to improve it:

- One addition that could be helpful to this section would be listing the vital signs you are referring to. Is it heart rate, blood pressure, oxygen saturation? More?

Response: We have added the specific vital signs (blood pressure, heart rate, respiratory rate, oxygen saturation, temperature, and blood glucose) in the Introduction to enhance clarity.

- Lines 66-68 are basically restating what is above and I don’t think they are needed at all. The introduction ended beautifully at line 65 on the previous page. If you really wanted a final sentence remove the repeated info and leave only “…Hospital. The results of this study will contribute evidence-based recommendations for optimizing stroke care in LMICs.”

Response: As advised, we have removed the redundant lines and retained only the essential summary sentence, which closes the Introduction more effectively.

Methods:

- Was ethical approval obtained from NIMR, the National IRB of Tanzania, in addition to the Research and Publication Committee of Muhimbili? I believe studies in Tanzania need NIMR approval, too.

Response: We appreciate the reviewer’s attention to ethical oversight. Ethical clearance for this study was obtained from the Research and Publication Committee of Muhimbili University of Health and Allied Sciences (MUHAS). As the study was not a randomized controlled trial and did not involve external collaborators, additional approval from the National Institute for Medical Research (NIMR) was not mandated under Tanzanian research ethics guidelines. This clarification has been added to the manuscript’s Ethics section.

- Were the abnormal vitals reported to clinicians in real time? (line 77-78) Consider adding this detail.

Response: Yes, all abnormal vital signs were promptly communicated to the clinical team. This has been explicitly stated in the Methods section under “Clinical data collection.”

- Lines 84-85. You do not need to re-state that studies indicated improved stroke outcomes as you already built the case well in the Introduction. Please remove.

Response: Agreed. This repetition has been removed for conciseness.

- Line 85. Do not say “we wanted to evaluate” but rather say “we evaluated” as it is more direct, crisp writing.

Response: Corrected as suggested.

- Line 91-92. Perhaps define stroke mimics and also reliable informant when describing exclusion criteria.

Response: Definitions have now been added to the Study population subsection of the Materials and methods section:

Stroke mimics were defined as clinical syndromes initially suspected to be stroke but later found to be due to other conditions (e.g., brain tumors, metabolic encephalopathies).

A reliable informant referred to a relative or caregiver able to provide accurate clinical and behavioral history for patients unable to communicate due to aphasia or impaired consciousness.

- In Sample Size, why was 35% relative reduction chosen?

Response: We thank the reviewer for this important observation. The expected 30-day mortality rates used in our sample size calculation—61% for the 12-hourly group and 37% for the 6-hourly group—were drawn from two prospective cohort studies conducted in similar sub-Saharan African settings (Tanzania and Nigeria, respectively). Comparing these two proportions yields an estimated 39% relative reduction in mortality.

However, to ensure a conservative and achievable effect size that remained clinically meaningful, we powered the study to detect at least a 35% relative reduction in mortality. This approach ensured that our sample size remained robust while grounded in real-world data, and allowed sufficient power to detect a mortality differen

---

## [Decision Letter · Decision Letter 1]

Intensive vital signs monitoring reduces 30-day mortality among stroke patients: A cohort study from Tanzania

PONE-D-25-10513R1

Dear Dr. Tumaini,

We’re pleased to inform you that your manuscript has been judged scientifically suitable for publication and will be formally accepted for publication once it meets all outstanding technical requirements.

Kind regards,

Elvan Wiyarta, M.D.

Academic Editor

PLOS ONE

Additional Editor Comments (optional):

Reviewers' comments:

Reviewer's Responses to Questions

**Comments to the Author**

1. If the authors have adequately addressed your comments raised in a previous round of review and you feel that this manuscript is now acceptable for publication, you may indicate that here to bypass the “Comments to the Author” section, enter your conflict of interest statement in the “Confidential to Editor” section, and submit your "Accept" recommendation.

Reviewer #1: All comments have been addressed

Reviewer #2: All comments have been addressed

2. Is the manuscript technically sound, and do the data support the conclusions?

Reviewer #1: Yes

Reviewer #2: (No Response)

3. Has the statistical analysis been performed appropriately and rigorously? 

Reviewer #1: Yes

Reviewer #2: (No Response)

4. Have the authors made all data underlying the findings in their manuscript fully available?

Reviewer #1: Yes

Reviewer #2: (No Response)

5. Is the manuscript presented in an intelligible fashion and written in standard English?

Reviewer #1: Yes

Reviewer #2: (No Response)

6. Review Comments to the Author

Reviewer #1: In the methods section: The reason why this is a cohort study has been justified to the reviewer in the reviewer comments section, and I am happy with that justification. This justification does not need to appear on the methods section of the paper (line 93 to 97 can be removed).

All other comments have been reasonably answered. congratulations to the team.

Reviewer #2: (No Response)

7. PLOS authors have the option to publish the peer review history of their article (what does this mean? ). If published, this will include your full peer review and any attached files.

**Do you want your identity to be public for this peer review?** For information about this choice, including consent withdrawal, please see our Privacy Policy .

Reviewer #1: No

Reviewer #2: No

---

## [Editor Report · Acceptance letter]

PONE-D-25-10513R1

PLOS ONE

Dear Dr. Tumaini,

I'm pleased to inform you that your manuscript has been deemed suitable for publication in PLOS ONE. Congratulations! Your manuscript is now being handed over to our production team.

Kind regards,

on behalf of

Mr. Elvan Wiyarta

Academic Editor

PLOS ONE